# THE INTRINSIC DIMENSION OF IMAGES AND ITS IMPACT ON LEARNING

**Phillip Pope[1], Chen Zhu[1], Ahmed Abdelkader[2], Micah Goldblum[1], Tom Goldstein[1]**
[1]Department of Computer Science, University of Maryland, College Park
[2]Oden Institute for Computational Engineering and Sciences, University of Texas at Austin
`{pepope,chenzhu}@umd.edu, akader@utexas.edu, {goldblum,tomg}@umd.edu`

## ABSTRACT

It is widely believed that natural image data exhibits low-dimensional structure despite the high dimensionality of conventional pixel representations. This idea underlies a common intuition for the remarkable success of deep learning in computer vision. In this work, we apply dimension estimation tools to popular datasets and investigate the role of low-dimensional structure in deep learning. We find that common natural image datasets indeed have very low intrinsic dimension relative to the high number of pixels in the images. Additionally, we find that low dimensional datasets are easier for neural networks to learn, and models solving these tasks generalize better from training to test data. Along the way, we develop a technique for validating our dimension estimation tools on synthetic data generated by GANs allowing us to actively manipulate the intrinsic dimension by controlling the image generation process. Code for our experiments may be found here.

## 1 INTRODUCTION

The idea that real-world data distributions can be described by very few variables underpins machine learning research from manifold learning to dimension reduction (Besold & Spokoiny, 2019; Fodor, 2002). The number of variables needed to describe a data distribution is known as its *intrinsic dimension* (ID). In applications, such as crystallography, computer graphics, and ecology, practitioners depend on data having low intrinsic dimension (Valle & Oganov, 2010; Desbrun et al., 2002; Laughlin, 2014). The utility of representations which are low-dimensional has motivated a variety of deep learning techniques including autoencoders and regularization methods (Hinton & Salakhutdinov, 2006; Vincent et al., 2010; Gonzalez & Balajewicz, 2018; Zhu et al., 2018).

It is also known that dimensionality plays a strong role in learning function approximations and non-linear class boundaries. The exponential cost of learning in high dimensions is easily captured by the trivial case of sampling a function on a cube; in $d$ dimensions, sampling only the cube vertices would require $2^d$ measurements. Similar behaviors emerge in learning theory. It is known that learning a manifold requires a number of samples that grows exponentially with the manifold's intrinsic dimension (Narayanan & Mitter, 2010). Similarly, the number of samples needed to learn a well-conditioned decision boundary between two classes is an exponential function of the intrinsic dimension of the manifold on which the classes lie (Narayanan & Niyogi, 2009). Furthermore, these learning bounds have no dependence on the ambient dimension in which manifold-structured datasets live.

In light of the exponentially large sample complexity of learning high-dimensional functions, the ability of neural networks to learn from image data is remarkable. Networks learn complex decision boundaries from small amounts of image data (often just a few hundred or thousand samples per class). At the same time, generative adversarial networks (GANs) are able to learn image "manifolds" from merely a few thousand samples. The seemingly low number of samples needed to learn these manifolds strongly suggests that image datasets have extremely low-dimensional structure.

Despite the established role of low dimensional data in deep learning, little is known about the intrinsic dimension of popular datasets and the impact of dimensionality on the performance of neural networks. Computational methods for estimating intrinsic dimension enable these measurements.

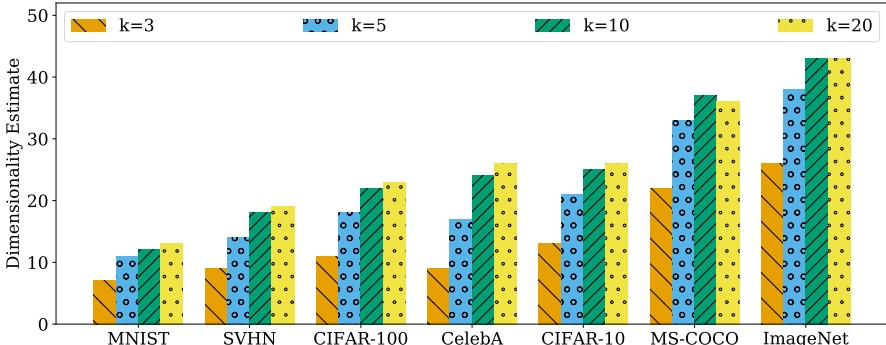

Figure 1: Estimates of the intrinsic dimension of commonly used datasets obtained using the MLE method with $k = 3, 5, 10, 20$ nearest neighbors (left to right). The trends are consistent using different $k$'s.

We adopt tools from the dimension estimation literature to shed light on dimensionality in settings of interest to the deep learning community. Our contributions can be summarized as follows:

- We verify the reliability of intrinsic dimension estimation on high-dimensional data using generative adversarial networks (GANs), a setting in which we can a priori upper-bound the intrinsic dimension of generated data by the dimension of the latent noise vector.

- We measure the dimensionality of popular datasets such as MNIST, CIFAR-10, and ImageNet. In our experiments, we find that natural image datasets whose images contain thousands of pixels can, in fact, be described by orders of magnitude fewer variables. For example, we estimate that ImageNet, despite containing $224 \times 224 \times 3 = 150528$ pixels per image, only has intrinsic dimension between 26 and 43; see Figure 1.

- We train classifiers on data, synthetic and real, of various intrinsic dimension and find that this variable correlates closely with the number of samples needed for learning. On the other hand, we find that extrinsic dimension, the dimension of the ambient space in which data is embedded, has little impact on generalization.

Together, these results put experimental weight behind the hypothesis that the unintuitively low dimensionality of natural images is being exploited by deep networks, and suggest that a characterization of this structure is an essential building block for a successful theory of deep learning.

## 2 RELATED WORK

While the hypothesis that natural images lie on or near a low-dimensional manifold is controversial, Goodfellow et al. (2016) argue that the low-dimensional manifold assumption is at least approximately correct for images, supported by two observations. First, natural images are locally connected, with each image surrounded by other highly similar images reachable through image transformations (e.g., contrast, brightness). Second, natural images seem to lie on a low-dimensional structure, as the probability distribution of images is highly concentrated; uniformly sampled pixels can hardly assemble a meaningful image. It is widely believed that the combination of natural scenes and sensor properties yields very sparse and concentrated image distributions, as has been supported by several empirical studies on image patches (Lee et al., 2003; Donoho & Grimes, 2005; Carlsson et al., 2008). This observation motivated work on efficient coding (Olshausen & Field, 1996) and served as a prior in computer vision (Peyré, 2009). Further, rigorous experiments have been conducted clearly supporting the low-dimensional manifold hypothesis for many image datasets (Ruderman, 1994; Schölkopf et al., 1998; Roweis & Saul, 2000; Tenenbaum et al., 2000; Brand, 2003); see also (Fefferman et al., 2016) for principled algorithms on verifying the manifold hypothesis.

The generalization literature seeks to understand why some models generalize better from training data to test data than others. One line of work suggests that the loss landscape geometry explains why neural networks generalize well (Huang et al., 2019). Other generalization work predicts that data with low dimension, along with other properties which do not include extrinsic dimension,

characterize the generalization difficulty of classification problems (Narayanan & Niyogi, 2009). In the context of deep learning, Gong et al. (2019) found that neural network features are low-dimensional. Ansuini et al. (2019) further found that the intrinsic dimension of features decreases in late layers of neural networks and observed interesting trends in the dimension of features in early layers. In contrast to Gong et al. (2019) and Ansuini et al. (2019), who find that the intrinsic dimension of internal representations is inversely correlated with high performance, we study the dimensionality of *data* and its impact on performance, and we make a similar finding. Zhu et al. (2018) proposed a regularizer derived from the intrinsic dimension of images augmented with their corresponding feature vectors. Another line of work in deep learning has found that neural networks rely heavily on textures which are low-dimensional (Geirhos et al., 2018; Brendel & Bethge, 2019). Similarly, some have suggested that natural images can be represented as mixtures of textures which lie on a low-dimensional manifold (Vacher & Coen-Cagli, 2019; Vacher et al., 2020).

## 3 INTRINSIC DIMENSION ESTIMATION

Given a set of sample points $\mathcal{P} \subset \mathbb{R}^N$, it is common to assume that $\mathcal{P}$ lies on or near a low-dimensional manifold $\mathcal{M} \subseteq \mathbb{R}^N$ of intrinsic dimension $\dim(\mathcal{M}) = d \ll N$. As a measure of the degrees of freedom in a dataset, as well as the information content, there is great interest in estimating the intrinsic dimension $d$. In the remainder of this section, we briefly describe the dimension estimation method we use in this paper; for further information, see (Kim et al., 2019) and references therein.

One of the main approaches to intrinsic dimension estimation is to examine a neighborhood around each point in the dataset, and compute the Euclidean distance to the $k^{th}$ nearest neighbor. Assuming that density is constant within small neighborhoods, the *Maximum Likelihood Estimation (MLE)* of Levina & Bickel (2005) uses a Poisson process to model the number of points found by random sampling within a given radius around each sample point. By relating the rate of this process to the surface area of the sphere, the likelihood equations yield an estimate of the ID at a given point $x$ as:

$$\hat{m}_k(x) = \left[ \frac{1}{k-1} \sum_{j=1}^{k-1} \log \frac{T_k(x)}{T_j(x)} \right]^{-1}, \tag{1}$$

where $T_j(x)$ is the Euclidean ($\ell_2$) distance from $x$ to its $j^{th}$ nearest neighbor. Levina & Bickel (2005) propose to average the local estimates at each point to obtain a global estimate $\bar{m}_k = \frac{1}{n}\sum_{i=1}^n \hat{m}_k(x_i)$. MacKay & Ghahramani (2005) suggestion a correction based on averaging of inverses

$$\bar{m}_k = \left[ \frac{1}{n} \sum_{i=1}^n \hat{m}_k(x_i)^{-1} \right]^{-1} = \left[ \frac{1}{n(k-1)} \sum_{i=1}^n \sum_{j=1}^{k-1} \log \frac{T_k(x_i)}{T_j(x_i)} \right]^{-1}, \tag{2}$$

where $n$ is the number of samples. We use Equation (2) as our MLE estimator throughout this paper.

Since the geometry of natural images is complex and unknown, we face two challenges when verifying the accuracy of MLE on natural image datasets. First, we need to choose a proper value of $k$. As shown by MacKay & Ghahramani (2005), the positive bias of the corrected estimator Equation (2) increases as $k$ increases, but the variance decreases. In order to navigate this bias-variance tradeoff, we try various values of $k$ in Section 4. Second, in addition to the aforementioned local uniformity assumption, MLE assumes that data arises as a sequence of i.i.d. random variables which can be written as a continuous and sufficiently smooth function of a random variable with smooth density, which may or may not be true for natural image datasets. While the truth of these assumptions is unknown on natural images, we verify the accuracy of our MLE estimates in a controlled setting in the following section.

We briefly discuss other notable techniques for dimensionality estimation. GeoMLE (Gomtsyan et al., 2019) attempts to account for non-uniformity of density and nonlinearity of manifold using a polynomial regression of standard MLE based on distances to nearest neighbors in different sized neighborhoods. However, GeoMLE chooses to approximate averages of $\hat{m}_k(x_i)$, instead of

averaging its reciprocal like Equation (2), resulting in a potentially wrong maximum likelihood estimator. As a result, we find its estimation deviates significantly from expected dimensionalities. TwoNN (Facco et al., 2017) is based on the ratio of the distances to the first and second nearest neighbors. Finally, the approach of (Granata & Carnevale, 2016) considers the distribution of geodesic distances over the data manifold, approximated by distances through kNN graphs, compared to the distribution of distances over hyperspheres of varying dimension. Unlike MLE, our preliminary experiments suggest that these techniques do not provide reasonable estimates for some natural and synthetic images which are key to this work; see Appendix A.5 for further discussion.

## 4    VALIDATING DIMENSION ESTIMATION WITH SYNTHETIC DATA

Dimensionality estimates are often applied on "simple" manifolds or toy datasets where the dimensionality is known, and so the accuracy of the methods can be validated. Image manifolds, by contrast, are highly complex, may contain many symmetries and modes, and are of unknown dimension. In principle, there is no reason why MLE-based dimensionality estimates cannot be applied to image datasets. However, because we lack knowledge of the exact dimensionality of image datasets, we cannot directly verify that MLE-based dimensionality estimates scale up to the complexity of image structures.

There is an inherent uncertainty in estimating the ID of a given dataset. First, we cannot be sure if the dataset actually resembles a sampling of points on or near a manifold. Second, there are typically no guarantees that the sampling satisfies the conditions assumed by the ID estimators we are using.

Towards a principled application of ID estimates in contexts of practical relevance to deep learning, we begin by validating that MLE methods can generate accurate dimensionality estimates for complex image data. We do this by generating synthetic image datasets using generative models for which the intrinsic dimensionality can be upper-bounded a priori. We believe such validations are essential to put recent findings in perspective (Gong et al., 2019; Ansuini et al., 2019).

**GAN Images**    We use the BigGAN variant with $128$ latent entries and outputs of size $128 \times 128 \times 3$ trained on the ImageNet dataset (Deng et al., 2009). Using this GAN, we generate datasets with a varying number of images, where we fix most entries of the latent vectors to zero leaving only $\bar{d}$ free entries to be chosen at random. As we increase the number of free entries, we expect the intrinsic dimension to increase with $\bar{d}$ as an upper bound; see Section A.1 for further discussion.

In particular, we create several synthetic datasets of varying intrinsic dimensionality using the ImageNet class, `basenji`, and check if the estimates match our expectation. As seen in Figure 2, we observe increasing diversity with increasing intrinsic dimension. In Figure 3, we show convergence of the MLE estimate on `basenji` data with dimension bounded above by $\bar{d} = 10$. We observe that the estimates can be sensitive to the choice of $k$ as discussed in prior work; see Appendix A.2 for additional GAN classes.

**Scaling to large datasets.**    We develop a practical approach for estimating the ID of large datasets such as ImageNet. In this approach, we randomly select a fraction $\alpha$ of the dataset as anchors. Then, we evaluate the MLE estimate using only the anchor points, where nearest-neighbors are computed over the entire dataset. Note that, when anchors are chosen randomly, this acceleration has no impact on the expected value of the result. See Appendix A.3 for an evaluation of this approach.

## 5    THE INTRINSIC DIMENSION OF POPULAR DATASETS

In this section, we measure the intrinsic dimensions of a number of popular datasets including MNIST (Deng, 2012), SVHN (Netzer et al., 2011), CIFAR-10 and CIFAR-100 (Krizhevsky et al., 2009), ImageNet (Deng et al., 2009), MS-COCO (Lin et al., 2014), and CelebA (Liu et al., 2015). Using three different parameter settings for the MLE ID estimator, we find that the ID is indeed much smaller than the number of pixels; see Table 5. Notice that the rank order of datasets by dimension does not depend on the choice of $k$. A comparison of state-of-the-art (SOTA) classification accuracy on each respective dataset[1] with the dimension estimates suggests a negative correlation

---

[1]Values from `https://paperswithcode.com/task/image-classification`.

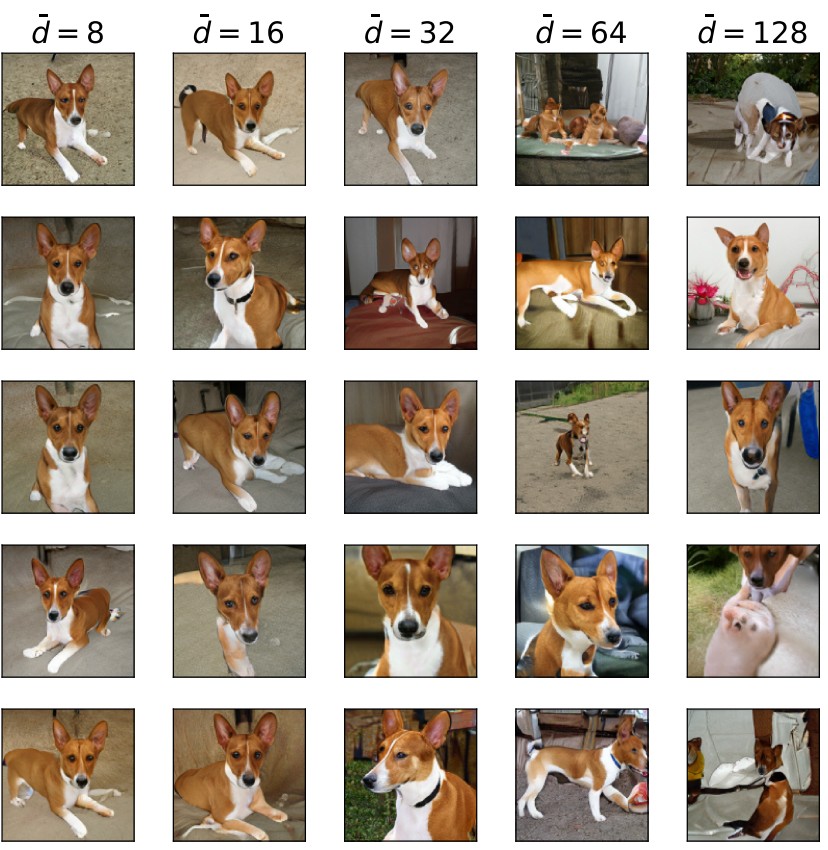

Figure 2: Visualization of `basenji` GAN samples of varying intrinsic dimension.

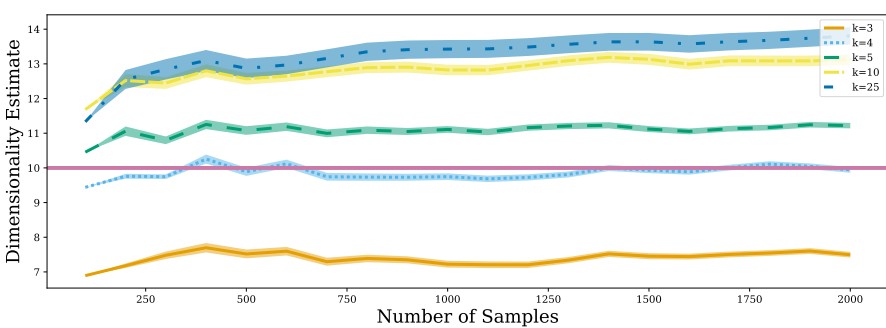

Figure 3: Validation of MLE estimate on synthetic `basenji` data with $\bar{d} = 10$ free entries. We observe the estimates to converge around the expected dimensionality of 10. Standard errors plotted with $N = 5$ replicates over random samples of the data.

between the intrinsic dimension and test accuracy. In the next section, we take a closer look at this phenomenon through a series of dedicated experiments.

# 6 INTRINSIC DIMENSION AND GENERALIZATION

Learning theory work has established that learning a manifold requires a number of samples that grows exponentially with the manifold's intrinsic dimension (Narayanan & Mitter, 2010), but the required number of samples is independent of the extrinsic dimension. Specifically, the number of

| Dataset | MNIST | SVHN | CIFAR-100 | CelebA | CIFAR-10 | MS-COCO | ImageNet |
|---|---|---|---|---|---|---|---|
| MLE ($k$=3) | 7 | 9 | 11 | 9 | 13 | 22 | 26 |
| MLE ($k$=5) | 11 | 14 | 18 | 17 | 21 | 33 | 38 |
| MLE ($k$=10) | 12 | 18 | 22 | 24 | 25 | 37 | 43 |
| MLE ($k$=20) | 13 | 19 | 23 | 26 | 26 | 36 | 43 |
| SOTA Accuracy | 99.84 | 99.01 | 93.51 | - | 99.37 | - | 88.55 |

Table 1: The MLE estimates for practical image datasets, and the state-of-the-art test-set image classification accuracy (for classification problems only) for these datasets.

samples needed to learn a well-conditioned decision boundary between two classes is exponential in the intrinsic dimension of the manifold on which the classes lie (Narayanan & Niyogi, 2009).

We leverage dimension estimation tools to empirically verify these theoretical findings using a family of binary classification problems defined over both synthetic and real datasets of varying intrinsic dimension. In these experiments, we observe a connection between the intrinsic dimension of data and generalization. Specifically, we find that classification problems on data of lower intrinsic dimensionality are easier to solve.

### 6.1 Synthetic GAN data: Sample complexity depends on intrinsic (not extrinsic) dimensionality

The synthetic GAN data generation technique described in Section 4 provides a unique opportunity to test the relationship between generalization and the intrinsic/extrinsic dimensionality of images. By creating datasets with controlled intrinsic dimensionality, we may compare their *sample complexity*, that is the number of samples required to obtain a given level of test error. Specifically we test the following two hypotheses (1) data of lower intrinsic dimensionality has lower sample complexity than that of higher intrinsic dimensionality and (2) extrinsic dimensionality is irrelevant for sample complexity.

To investigate hypothesis (1), we create four synthetic datasets of varying intrinsic dimensionality: $16, 32, 64, 128$, *fixed* extrinsic dimensionality: $3 \times 32 \times 32$, and two classes: `basenji` and `beagle`. For each dataset we fix a test set of size $N = 1700$. For all experiments, we use the `ResNet-18` (width = 64) architecture (He et al., 2016). We then train models until they fit their entire training set with increasing amounts of training samples and measure the test error. We show these results in Figure 4. Observing the varying rates of growth, we see that data of higher intrinsic dimension requires more samples to achieve a given test error.

For hypothesis (2), we carry out the same experiment with the roles of intrinsic and extrinsic dimension switched. We create four synthetic datasets of varying *extrinsic* dimensionality by resizing the images with nearest-neighbor interpolation. Specifically we create 6 datasets of square, 3-channel images of sizes $16, 32, 64, 128, 256$, *fixed* intrinsic dimensionality of size $128$, and all other experimental details the same. We show these results in Figure 5. Observing the lack of variable growth rates, we see that extrinsic dimension has little to no effect on sample complexity.

To the best of our knowledge, this is the first experimental demonstration that *intrinsic but not extrinsic dimensionality matters for the generalization of deep networks*.

### 6.2 Real Data: Intrinsic dimensionality matters for generalization

Next, we examine the sample complexity of binary classification tasks from four common image datasets: MNIST, SVHN, CIFAR-10, and ImageNet. This case differs from the synthetic case in that we have no control over each dataset's intrinsic dimension. Instead, we estimate it via the MLE method discussed in Section 3. To account for variable difficulty of classes, we randomly sample 5 class pairs from each dataset and run the previously described sample complexity experiment. Note that these subsets differ from those used in Table 5, where the estimates are taken from the entire dataset and across all classes.

On these sampled subsets, we find the MLE estimates as shown in Table 2. Note that these estimates are consistent with expectation, e.g. MNIST is qualitatively simpler then SVHN or CIFAR-10.

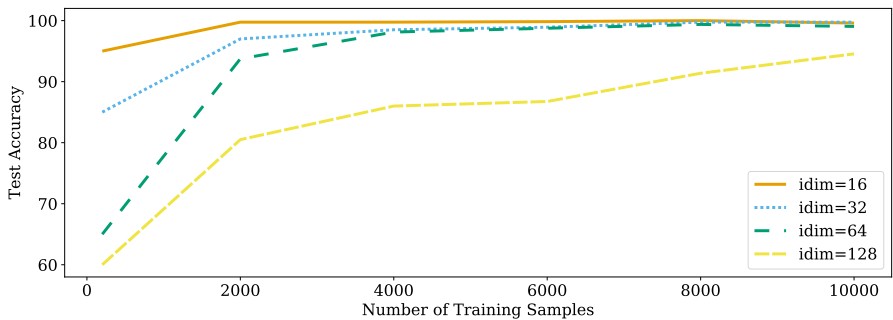

Figure 4: Sample complexity of synthetic datasets of varying intrinsic dimensionality.

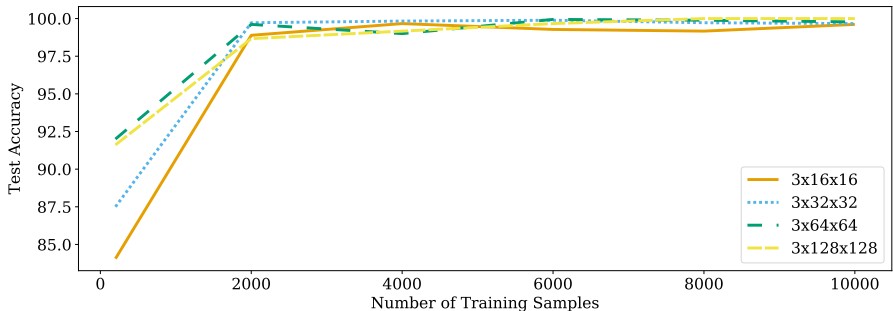

Figure 5: Sample complexity of synthetic datasets of varying extrinsic dimensionality.

We conduct the same sample complexity experiment as the previous section on the datasets. Because these datasets are ordinarily of varying extrinsic dimensionality, we resize all to size $32 \times 32 \times 3$ (before applying MLE). We report results in Figure 6, where we overall observe trends ordered by intrinsic dimensionality estimate. These results are consistent with expectation of the relative hardness of each dataset. However, there are some notable differences from the synthetic case. Several unexpected cross-over points exist in the low-sample regime, and the gap between SVHN and CIFAR-10 is smaller than one may expect based on their estimated intrinsic dimension.

From these observations we conclude that intrinsic dimensionality is indeed relevant to generalization on real data, but it is not the only feature of data that influences sample complexity.

|  | MNIST | SVHN | CIFAR-10 | ImageNet |
|---|---|---|---|---|
| $k = 3$ | 7.5 (0.2) | 8.5 (0.1) | 11.4 (0.2) | 15.4 (0.8) |
| $k = 4$ | 9.8 (0.3) | 11.6 (0.1) | 15.9 (0.2) | 19.8 (0.9) |
| $k = 5$ | 10.9 (0.4) | 13.2 (0.1) | 18.3 (0.3) | 21.6 (1.0) |

Table 2: Mean and standard error of estimated intrinsic dimensions for practical datasets under the same resolution $32 \times 32 \times 3$ using different $k$'s. These results are consistent with the test accuracies on these datasets under the same resolution.

## 6.3 REAL DATA: ADDING NOISE CHANGES DIMENSIONALITY TO AFFECT GENERALIZATION

In this section, we examine an alternative technique for changing the intrinsic dimension of a *real* dataset: adding noise to images. Here we leverage the fact that uniformly sampled noise in $[0, 1]^{\underline{d}}$ has dimension $\underline{d}$. We thus add independent noise, drawn uniformly from a fixed randomly oriented $\underline{d}$-dimensional unit hypercube embedded in pixel space, to each sample in a dataset. This procedure ensures that the dataset has dimension at least $\underline{d}$. Since the natural data we use has low dimension, and the hypercubes have high dimension, this procedure specifically increases dimensionality. We note that estimation error may occur when there is an insufficient number of samples to achieve

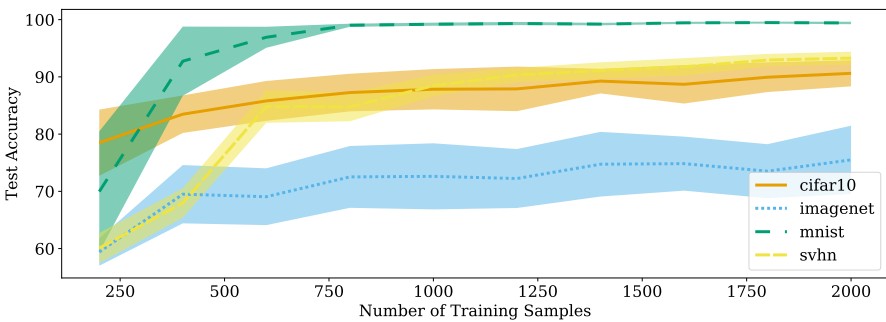

Figure 6: Sample complexity of real datasets. Standard errors are shown $N = 5$ class pairs.

a proper estimate. Since the variation in images in a dataset may still be dominated by non-noise directions, we expect to underestimate the new increased dimensions of these noised datasets.

Starting with CIFAR-10 data, we add noise of varying dimensions, where we replace pixels at random in the image. We only add noise to an image once to keep the augmented dataset the same size as the original. We use the following noise dimensionalities: $256, 512, 1024, 2048, 2560$. The estimated dimensions of the noised datasets are listed in Table 3. We see that intrinsic dimension increases with increasing noise dimensionality, but dimensionality does not saturate to the maximum true dimension, likely due to a poverty of samples.

On these noisy CIFAR-10 datasets, we again carry out the sample complexity experiment of the previous sections. We show results in Figure 7. We observe sample complexity largely in the same order as intrinsic dimension.

|  | $\underline{d} = 256$ | $\underline{d} = 512$ | $\underline{d} = 1024$ | $\underline{d} = 1536$ | $\underline{d} = 2048$ | $\underline{d} = 2560$ |
|---|---|---|---|---|---|---|
| $k = 3$ | 19.7 | 30.9 | 57.1 | 77.8 | 110.0 | 136.1 |
| $k = 4$ | 25.2 | 39.1 | 72.8 | 101.3 | 142.1 | 177.7 |
| $k = 5$ | 27.6 | 42.5 | 78.3 | 110.2 | 153.4 | 196.6 |

Table 3: Estimated intrinsic dimension of the CIFAR-10 dataset after adding different dimensions ($d$) of uniformly sampled noise using different $k$'s. The estimated dimension consistently increases with $d$ under different $k$'s.

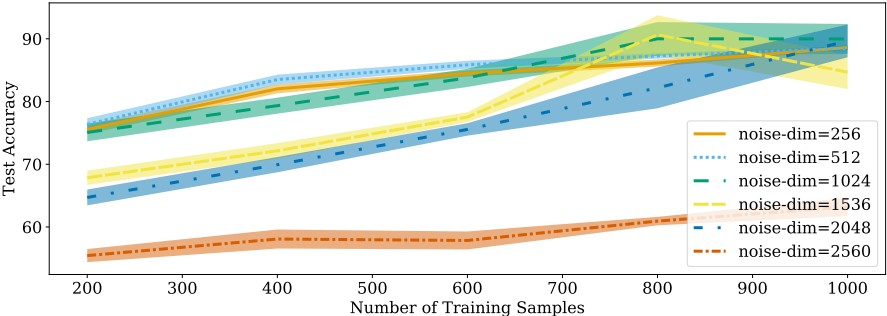

Figure 7: Sample complexity of noisy datasets. Standard errors are shown $N = 5$ random subsets of the data.

### 6.4 MANIPULATING THE INTRINSIC DIMENSIONALITY OF FONTS

In this section, we describe a final technique for studying the effect of intrinsic dimensionality on sample complexity on the recently proposed FONTS dataset (Stutz et al., 2019). Beginning with a collection of characters and font types, termed a *prototype* set by the authors, FONTS datasets are constructed using a fixed set of data augmentations: scaling, translation, rotation, and sheering. In

principle, these augmentations each increase the intrinsic dimension of the prototype set allowing us to synthetically alter the intrinsic dimension by varying the number of augmentations used.

We construct 5 FONTS datasets in this way, FONTS-$\{0, 1, 2, 3, 4\}$, where the suffix denotes the number of transformations used in the data generation process. The MLE estimates on each of the datasets are given in Table 4.

Consistent with expectation, we observe that MLE methods consistently resolve the increased dimensionality of transformed datasets. Carrying out the sample complexity experiment of the previous section, we report results in Figure 8. We observe *again* that, on the whole, sample complexity is ordered by intrinsic dimension.

|         | FONTS-0   | FONTS-1   | FONTS-2   | FONTS-3   | FONTS-4   |
|---------|-----------|-----------|-----------|-----------|-----------|
| $k = 3$ | 1.8 (0.1) | 3.8 (0.1) | 5.1 (0.2) | 5.8 (0.2) | 6.1 (0.3) |
| $k = 4$ | 2.7 (0.1) | 5.2 (0.1) | 6.9 (0.2) | 7.8 (0.3) | 8.3 (0.4) |
| $k = 5$ | 3.2 (0.1) | 5.9 (0.1) | 7.8 (0.3) | 8.8 (0.4) | 9.4 (0.4) |

Table 4: Mean and standard errror of MLE estimates on the FONTS dataset under different number of transforms using different $k$'s. Again, the ranks of the estimated IDs are consistent, and the estimated IDs increase with the number of transforms.

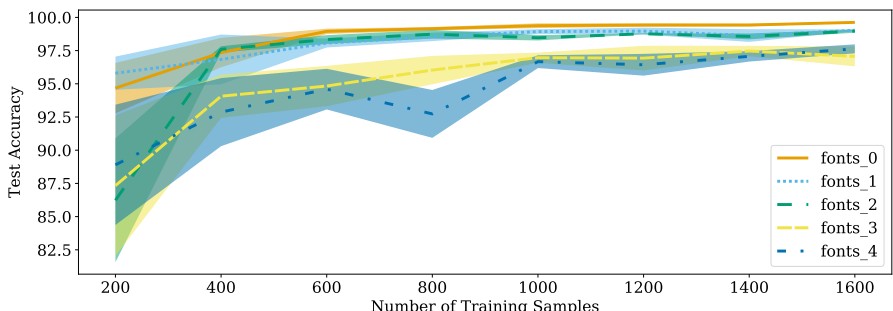

Figure 8: Sample complexity of FONTS datasets.

# 7 DISCUSSION

In this work, we measure the intrinsic dimension of popular image datasets and show that the intrinsic dimension of data matters for deep learning. While there may be many factors, such as class separation and the number of classes, which determine generalization, we build the case that intrinsic dimension is one of these important factors. Along the way, we introduce a technique for using GANs to synthesize data while manipulating dimensionality. This technique is useful not only for validating dimension estimation methods but also for examining the learning behavior of neural networks under a dimension-controlled environment. In addition to synthetic data, we verify that dimension plays a large role in learning on natural data. Our results support the commonly held belief that low dimensional structure underpins the success of deep learning on high-resolution data.

These findings raise a number of salient directions for future work. Methods for enhancing neural network learning on high-dimensional data could improve generalization on hard vision problems. To this end, a deeper understanding of the mechanisms by which dimensionality plays a role in learning may enable such methods. Additionally, our work indicates that different dimensionality estimation tools are better suited for different settings. Future work on computing tighter and more reliable estimates specific to image data would allow the community to more precisely study the relationship between the dimensionality of image datasets and learning.

ACKNOWLEDGEMENTS

This work was supported by the DARPA GARD and DARPA QED programs. Further support was provided by the AFOSR MURI program, and the National Science Foundation's DMS division. Computation resources were funded by the Sloan Foundation.

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

# A    VALIDATION OF ID ESTIMATES

In this section, we present additional discussion results and discussion relevant to the ID estimation and related validation experiments in Section 4.

## A.1    GAN PROPERTIES

We devise a method for validating ID measurements in a controlled setting using images generated by GANs. To justify this method, we first note that the image of $\mathbb{R}^d$ under a locally Lipschitz function can be a manifold with dimension at most $d$. Then, consider that the BigGAN generator, a convolutional neural network with ReLU activations, is a function with this property (Brock et al., 2018).

Specifically, BigGAN can be written as a composition of linear functions, translations, and ReLU activation functions. Individually, these operations do not increase dimension, and by a composition property, their composition cannot increase dimensionality either. The more general fact that the image of $\mathbb{R}^d$ under a locally Lipschitz function can be a manifold with dimension at most $d$ follows from Sard's theorem (Sard, 1942).

## A.2    CONVERGENCE FOR MORE GAN CLASSES

We include additional results on the estimation of ID for synthetic GAN images from various ImageNet classes with $\bar{d} = 10$ free entries out of the 128-dimensional latent vector input to the GAN; see Figures 10 and 11 below. As observed earlier in Section 4, the MLE estimates are sensitive to the choice of $k$, where we expect the ID to be close to 10 given the way we sample the latent vectors to use for the GAN. We note that for a number of classes, all choices of $k$ we considered seem to underestimate the ID.

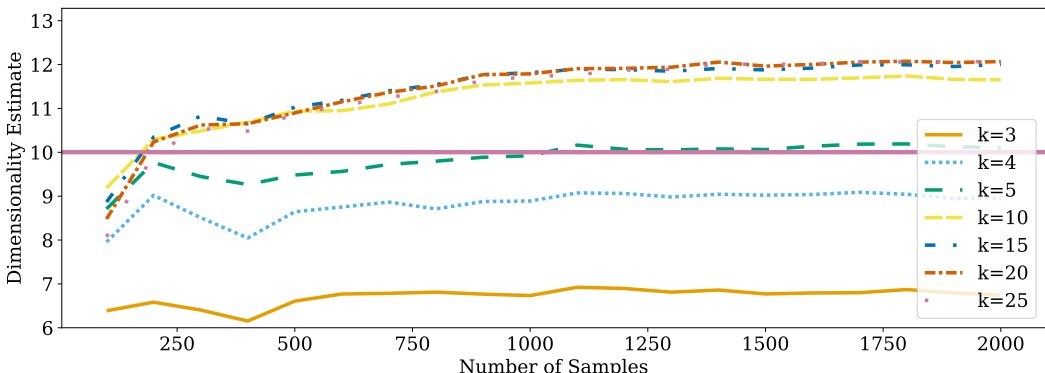

Figure 9:   Validation of MLE estimates on synthetic `daisy` data with $\bar{d} = 10$.

## A.3    SUBSAMPLING FOR LARGE DATASETS

In Figure 12 we validate the anchor approximation on `basenji` data of dimension 10 for varying anchor ratio $\alpha$. Then, in Figure 13 we validate the anchor approximation on `tree-frog` data with $\bar{d} = 32$ for varying $k$ while fixing the anchor ratio at $\alpha = 0.001$.

## A.4    RELATIONSHIP BETWEEN $k$ IN THE MLE METHOD AND DATASET DIMENSION

The choice of $k$ may affect the dimensionality estimates. To better understand this relationship, we conducted additional studies for the MLE method using synthetic GAN data generated as described in Section 4. Table 5 shows the estimation results for various $k$ and `basenji` data with $\bar{d} \in \{2, 4, 8, 16, 32, 64, 128\}$. We use a fixed number of samples ($n = 10000$). We observe that the estimated intrinsic dimension increases with $k$, and large values can yield overestimates for ID. These results agree with the work of Levina & Bickel (2005) who suggest that low values of $k$ may

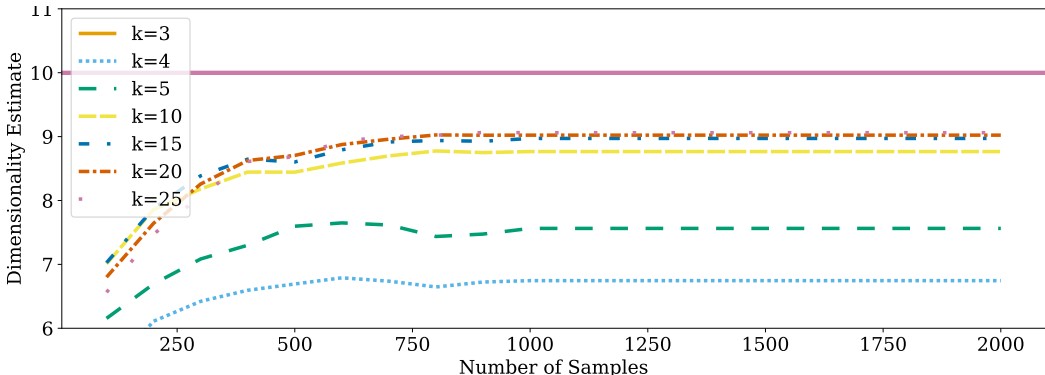

Figure 10: Validation of MLE estimate on synthetic `soap-bubbles` data with $\bar{d} = 10$.

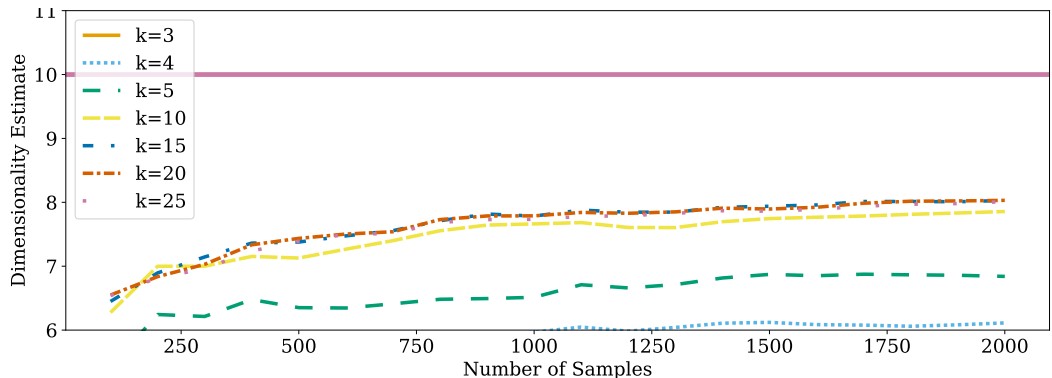

Figure 11: Validation of MLE estimate on synthetic `coffee` data with 10 free entries. Note that the estimates do not converge around the upper bound of $\bar{d} = 10$, which suggests that data generated from this class is not of full dimension.

produce an estimator with higher variance while higher values of $k$ produce an estimator with higher positive bias. Given that we have access to large amounts of synthetic data, we opt for lower values of $k$ in our work. We do not choose a particular value of $k$, and we report all experiments with multiple values.

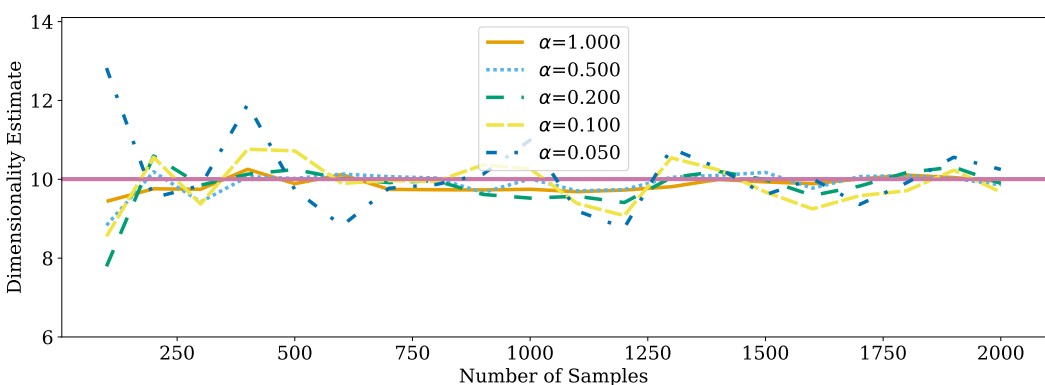

Figure 12: Validation of anchor approximation on `basenji` with $\bar{d} = 10$.

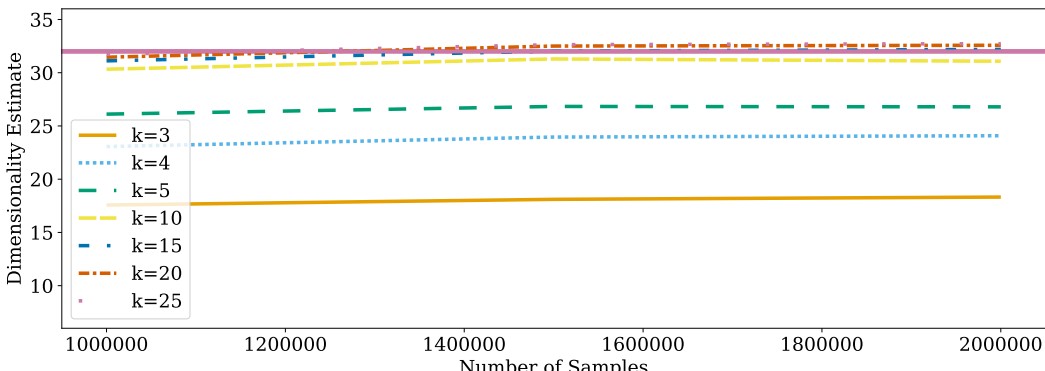

Figure 13: Validation of anchor approximation on `tree-frog` with $\bar{d} = 32$ and $\alpha = 0.001$.

| $k$ | $\bar{d}$ | | | | | | |
|---|---|---|---|---|---|---|---|
| | 2 | 4 | 8 | 16 | 32 | 64 | 128 |
| 3 | 1.1 | 2.6 | 6.1 | 10.5 | 16.0 | 20.0 | 20.0 |
| 4 | 1.5 | 3.6 | 8.2 | 14.0 | 21.0 | 26.0 | 26.0 |
| 5 | 1.7 | 4.1 | 9.3 | 15.7 | 23.5 | 28.7 | 28.5 |
| 6 | 1.8 | 4.4 | 9.9 | 16.6 | 24.9 | 30.3 | 29.9 |
| 7 | 1.9 | 4.6 | 10.4 | 17.2 | 25.8 | 31.2 | 30.6 |
| 8 | 1.9 | 4.7 | 10.7 | 17.6 | 26.4 | 31.7 | 31.1 |
| 9 | 2.0 | 4.9 | 10.9 | 18.0 | 26.8 | 31.9 | 31.5 |
| 10 | 2.0 | 5.0 | 11.1 | 18.2 | 27.1 | 32.1 | 31.7 |
| 15 | 2.1 | 5.3 | 11.6 | 18.8 | 27.8 | 32.3 | 31.7 |
| 20 | 2.2 | 5.5 | 11.8 | 19.0 | 27.9 | 31.9 | 31.3 |
| 25 | 2.2 | 5.7 | 12.0 | 19.2 | 27.9 | 31.5 | 30.8 |

Table 5: Additional results on the relationship between $k$ and MLE dimensionality estimate for synthetic `basenji` images with varying $\bar{d}$.

## A.5    COMPARING MLE TO OTHER ESTIMATORS IN A CONTROLLED SETTING

To validate MLE in comparison to other estimators, we evaluate three other dimensionality estimation methods: GeoMLE (Gomtsyan et al., 2019), TwoNN (Facco et al., 2017) and kNN graph distances (Granata & Carnevale, 2016). For GeoMLE, we sample a total of 20 bootstrap subsets ($M = 20$) and use $k_1 = 20, k_2 = 55$ as recommended by Gomtsyan et al. (2019). To extend the kNN graph distance method to large datasets, we randomly sample a subset of samples (fixed to 10,000 for datasets with more than 10,000 samples) and use shortest graph distances to their $k$ nearest neighbors to estimate the IDs. Other settings are as default in the implementations of Granata & Carnevale (2016).

First, we validate each method on datasets of uniformly sampled from $d$-dimensional hypercubes. We report these results in Figure 14. Each method works reasonable on low-dimensional cubes. We observed the Shortest Path method to give erratic estimates on cubes of higher dimension, and have omitted these. TwoNN has poor sample efficiency for higher dimensional cubes. Interestingly, GeoMLE estimates these high dimensional cubes well.

Next, we report estimation results on `basenji_10` for TwoNN, kNN graph distance, and GeoMLE methods in Figure 15. Comparing against the MLE results in Figure 3, we observe that each other method does not achieve an accurate estimate *in this sample regime*, thus motivating our focus on MLE. Notably, GeoMLE and TwoNN severely overestimate dimension, while the kNN graph distance method severely underestimates dimension.

On MNIST, CIFAR-10, CIFAR-100, and SVHN, the results of these other estimation methods also deviate from expectation (Table 6). For example, TwoNN assigns a significantly higher dimension

estimate to MNIST than to CIFAR-100, which contradicts both intuition and the results of other estimators. We set $k = 4$ and number of bins to 1000, and use default settings for all other parameters including $r_{\text{MAX}}$.

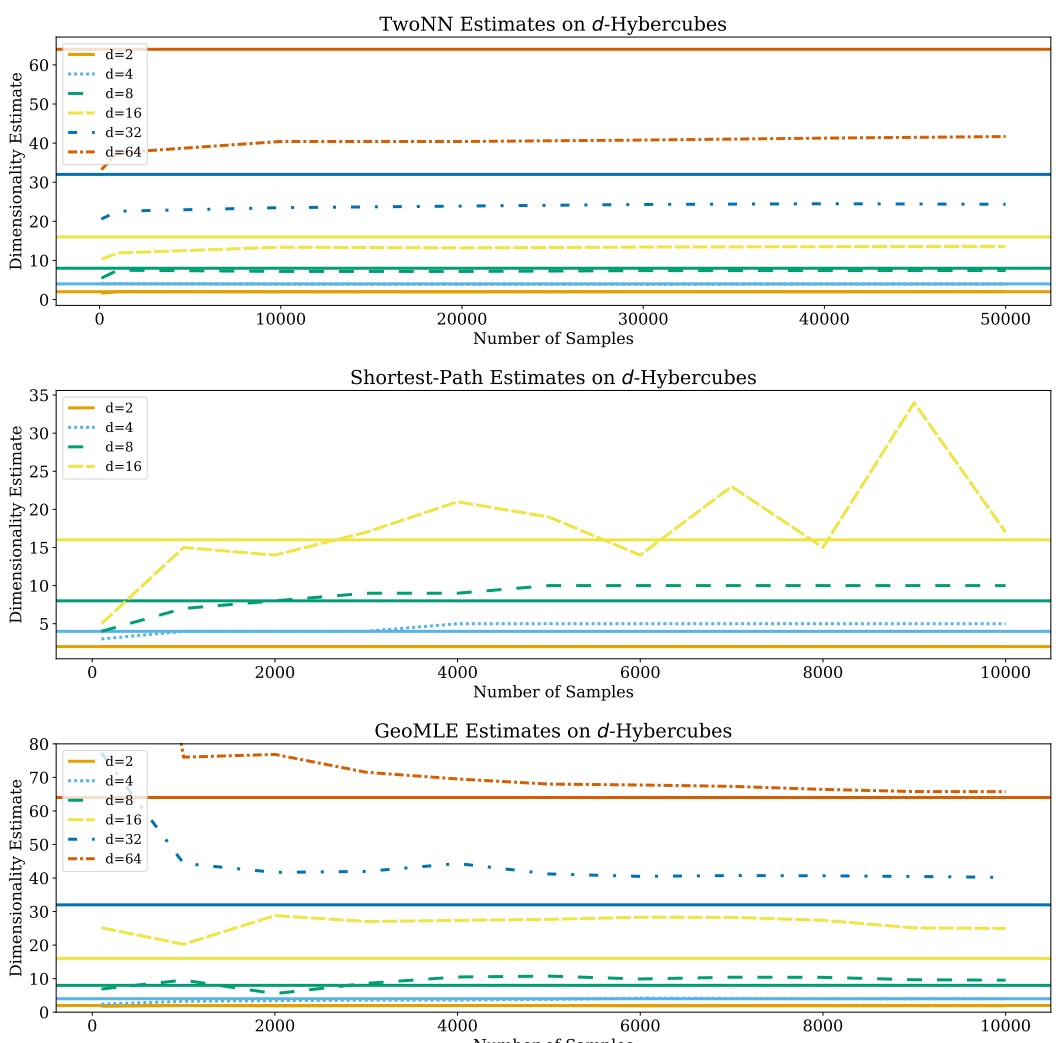

Figure 14: The TwoNN, Shortest-Path, and GeoMLE methods on $d$-dimensional Hypercube data. Each method estimates low-dimensional cubes well, validating their implementation.

| Dataset | MNIST | CIFAR-10 | CIFAR-100 | SVHN |
|---|---|---|---|---|
| MLE ($k = 5$) | 11 | 21 | 18 | 14 |
| GeoMLE ($k_1 = 20, k_2 = 55$) | 25 | 96 | 93 | 21 |
| TwoNN | 15 | 11 | 9 | 7 |
| kNN Graph Distance | 7 | 7 | 8 | 6 |

Table 6: Additional ID estimators on popular datasets.

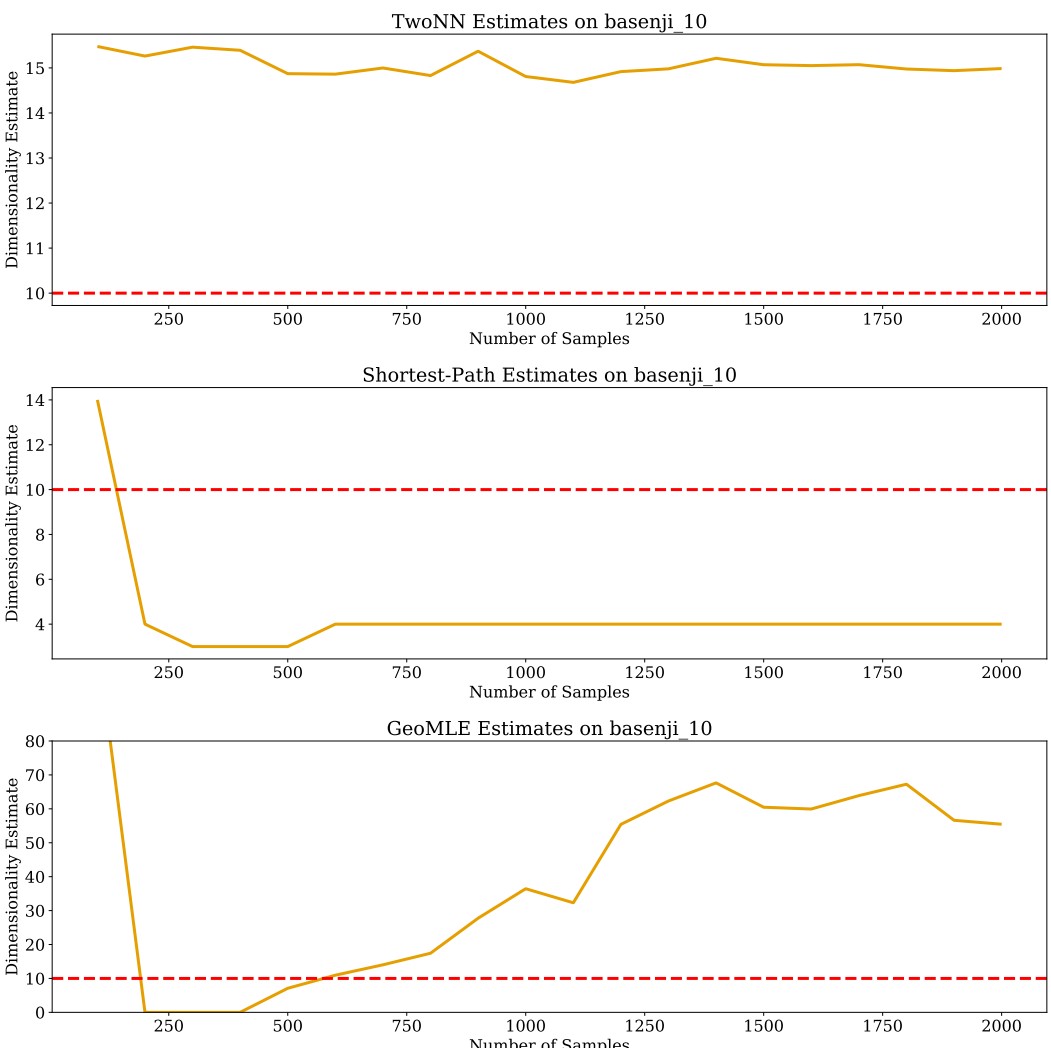

Figure 15: The TwoNN, Shortest-Path, and GeoMLE methods on basenji_10 data. The estimates do not converge around the expected value of $\bar{d} = 10$ in this sample regime.

