# OpenReview forum: "The Intrinsic Dimension of Images and Its Impact on Learning"
_ICLR.cc/2021/Conference — ICLR 2021 Spotlight_

### Official Review · AnonReviewer2 · 2020-10-20
**The authors report a novel application of GANs to validate the maximum likelihood estimator (MLE) of the intrinsic dimension (ID) of image data sets. Then they use the MLE ID estimator to characterize the intrinsic dimension of several commonly used computer vision data sets, and link the data set ID to the generalizability of trained classifiers. Also, they verify that dimension plays a large role in learning on natural data.**

**Rating:** 6
**Confidence:** 3

**Review:**


The authors report a novel application of GANs to validate the maximum likelihood estimator (MLE) of the intrinsic dimension (ID) of image data sets. Then they use the MLE ID estimator to characterize the intrinsic dimension of several commonly used computer vision data sets, and link the data set ID to the generalizability of trained classifiers.  They provide additional experiments that support the notion that it is intrinsic dimension, and not extrinsic dimension (i.e. # of pixels), that governs the performance of a binary classifier on these data sets. Also, they verify that dimension plays a large role in learning on natural data.

I found the paper to be clearly written, with only a few minor typographical errors in the writing, and the subject to be of practical usefulness to the deep learning community.  However, I do feel that the authors should perform a few additional experiments (I think they are reasonably simple) to improve the understanding of the results. I think this might be a topic of great interest to the computer vision community since this paper describes a novel application of GANs to study the sample complexity of convolutional neural networks,



*Review
Pros include:
innovative use of GANs to generate synthetic data of bounded intrinsic dimension
well written and easy to read
coherent story

Cons:
limited analysis and discussion of the role of the image class (particularly in ImageNet) on the MLE estimate of intrinsic dimension
Some details not listed in the paper, including the metric for the distance between images used to compute the MLE ID estimate
Non-sequitur in the analysis of the role of image extrinsic dimension in classifier generalization; see comments below.


In section 5.1, the authors use the method of image reshaping using nearest-neighbour interpolation to increase extrinsic dimensionality of images. A comparison of the generalization performance of classifier models on data sets with different extrinsic dimensionality is thus made.

>>The authors make the claim that generalization performance depends on intrinsic, but not extrinsic, the dimensionality of images. However, the method of image reshaping using interpolation seems to be reversible (i.e. lossless), and thus their generated images should have identical information content independent of extrinsic dimension.  For example, images collected using a high-resolution camera can not be faithfully reproduced by interpolation of images taken by a low-resolution camera.  It would be more interesting to consider the realistic scenario where images of lower dimension are generated by lossy downsampling of higher resolution images and to then characterize whether either generalization performance or intrinsic dimension were related to extrinsic dimension. I would guess that the intrinsic dimension of natural images is actually higher when they have a higher extrinsic dimension since there might be more fine details captured within images of high resolution.  Conceptual example: I might argue that the number and location of wrinkles on people’s faces is a variable that increases the intrinsic dimension of a facial photo data set, but only if the images are sufficiently high resolution to see the wrinkles.

In section 5.2, the authors downsample images of 5 randomly chosen class pairs from each real-world data set and then 1) compute MLE estimates of ID for each data set, and 2) compare the sample complexity of a binary classification problem on each.

>>It is interesting that while ImageNet MLE ID (k=5) is 38 and CIFAR-10 MLE ID (k=5) is 21, the MLE ID estimates (which are made using k=3 instead of k=5 or another previously used value; a choice not explained by the authors) for the low-resolution 5-random-class-pair samples of each data set are 15.4 and 11.4, respectively.  It seems that the ID of the two datasets are much more similar after the resolution-reduction and class-sampling, suggesting that intrinsic dimension of image data sets is strongly determined by either extrinsic dimension or the number (and kind) of classes present. The authors should comment on this fact.  Furthermore, given these observations, the authors should make some effort to determine which process (i.e., the lossy image reshaping, or the random sampling of 5 class pairs) contributes the most to the apparently much greater reduction in MLE ID that ImageNet suffers compared to CIFAR-10 in this experiment, and whether there are specific classes in ImageNet that contribute more to the intrinsic dimension of the data set than others. Perhaps the authors could simply compute the MLE ID estimate for the different classes of ImageNet (with and without image reshaping to 32x32x3), and then state which 5 class pairs were randomly chosen for the experiment in section 5.2.   These additional computations might explain the “unexpected cross-over points” mentioned in the paper.

In sections 5.3 and 5.4, the authors perform additional experiments showing that the intrinsic dimension (in this case modulated by either adding fixed-dimensionality noise or by applying image augmentation techniques) is associated with generalization performance of deep learning models.

>>In the discussion, the authors write “While there may be many factors, such as class separation and the number of classes, which determine generalization, we build the case that intrinsic dimension is one of these important factors.“  I think it might be true the intrinsic dimension is actually determined by both extrinsic dimensions and by inter-class and intra-class image diversity (i.e., number of classes and varying hardness of each class during classification). As I mentioned above, the authors could perform several simple experiments using their existing experimental framework to test this hypothesis, and I think doing so would significantly improve the quality of the paper.  Despite this point, I find the paper to be of acceptable quality.

---

> ### Author Response · Authors · 2020-11-24
> **Thank you for the insightful feedback.**
>
> We appreciate your feedback.  We have addressed each of your comments below.
>
> *“Some details not listed in the paper”*
>
> Thanks for pointing this out.  We have provided further clarification in our current draft.
>
> ---
>
> *“Method of image reshaping using interpolation seems to be reversible… For example, images collected using a high-resolution camera can not be faithfully reproduced... ”*
>
> We agree that upsizing images does not change the information content .  Anytime you embed a low-dimensional manifold into a high-dimensional space the problem does not become more difficult in a purely information theoretic sense.  However, we don’t think that we should take for granted that a neural net will perform the same on data with more pixels.  The network still needs to identify the relevant low-dimensional space on which the dataset resides, and one may think that this task becomes more complex when the dataset is contaminated with thousands of “nuisance” dimensions.   The network still succeeds though, and the high extrinsic dimensionality appears to be irrelevant.
>
> ---
>
> *“... lossy downsampling of higher resolution images … whether either generalization performance or intrinsic dimension were related to extrinsic dimension.”*
>
> We agree that higher resolution images with more detail have higher intrinsic dimension, but our goal was to verify the idea that the dimension of the pixel space in which the same image manifold is embedded does not affect the learnability.  To achieve this, we do a controlled experiment in which we upsample images to increase their extrinsic dimensions (dimension of the pixel space) while preserving intrinsic dimension. Downsampling images has the effect of changing both intrinsic and extrinsic dimension at the same time.  An interesting future direction could study how the type of camera sensor and resolution impact intrinsic dimensionality and learning.  However our experiments were designed to disentangle and individually study each factor.
>
> ---
>
> *“using k=3 instead of k=5 or another previously used value”*
>
> Thanks for pointing this out. We have now clarified our choices for k in the current draft (see Appendix A.4), and we have added measurements at multiple values of k for each experiment.  Now, we do not pick a single value of k for individual experiments; instead, we use multiple values of k for each experiment, and show each.  The trends we find do not depend on the choice of k.
>
> ---
>
> *“ID … more similar after the resolution-reduction ... intrinsic dimension of image data sets is strongly determined by extrinsic dimension.”*
>
> It is true that downsampling is lossy (as you mentioned above), and much of the high dimensionality of ImageNet is contained in high-frequency information that disappears under downsampling.  Since ImageNet is of much much higher extrinsic dimension than CIFAR-10, downsampling both to the same number of pixels does bring their intrinsic dimensions closer together, but the ID of ImageNet still remains higher.
>
> ---
>
> *“whether there are specific classes in ImageNet that contribute more to the intrinsic dimension.”*
>
> This is an interesting point.  We have now added a histogram of the ID estimates for different ImageNet classes.  Unfortunately, ImageNet does not contain sufficient data per class to obtain high quality estimates on a per-class basis.
>
> ---
> *“might explain the “unexpected cross-over points” mentioned in the paper.”*
>
> We agree that your explanation is plausible, and we did look into this phenomenon as you mentioned, but we were not able to explain the cross-over points.

---

### Official Review · AnonReviewer4 · 2020-10-31
**Interesting work**

**Rating:** 8
**Confidence:** 4

**Review:**

## Review

### Summary

The paper proposes an empirical analysis of the dimension of natural images of multiple datasets.
The contributions are:
1. A validation for nat. im. of previously proposed dimension estimation methods (using GAN to control the intrinsic dimension of the generated im.)
2. A confirmation that intrinsic dimension of nat. im. is lower than the dimension of their pixel space
3. That the lower the intrinsic dimension the easier the learning (for neural net)

### Strengths

* The paper is well-written and easy to follow.
* The data analysis pipeline is convincing.
* Up to my knowledge there is no such a clear statement about the dimension of natural image

### Weaknesses

* The results are not sufficiently discussed. I think the idea that nat. im. are low-d is more controversial than it is presented. It is sometime proposed that image patches (which are more likely to be textures) are low-d (eg Brendel & Bethge ICLR 2019). In relation, to neural net learning, these are known to be biased toward textures (Geirhos et. al. ICLR 2019). So among your contribution 3/ can be true while 2/ is not (but then what would explain your finding ?). I mean that in fact 3/ is more due two the low-d of textures than the low-d of nat. im. (which would still be too high).Complementary to this, it is suggested that natural images can be viewed as mixture of textures which belong to different low-d manifold (Vacher & Coen-Cagli, arXiv 1905.10629; Vacher et. al., NeurIPS 2020).

### Minor comments

* None

---

> ### Author Response · Authors · 2020-11-24
> **Thank you for your interest and feedback.**
>
> Thank you for your interest.  We agree that the controversy surrounding the dimensionality of natural images should have been emphasized, and we have thus updated our related works to reflect additional literature on this subject.
>
> The hypothesis that neural networks, which rely heavily on textures, learn well on natural images as a result of low dimensional textures (despite the full image space having high dimension) is an interesting topic for future work.  While this is not the focus of our paper, there has indeed been terrific work on image textures and their role in learning.  Our GAN experiments do not allow us to control the dimensionality of image patches and textures, but there may be other ways of controlling these using, for example, Fourier filters.  We have added citations to the works you mentioned to encourage our readers to explore these directions further.

---

### Official Review · AnonReviewer3 · 2020-11-01
**Interesting work on dimension of images but the development can be more solid**

**Rating:** 7
**Confidence:** 4

**Review:**

This paper studies the intrinsic dimension of image datasets and connects it to the generalization ability of deep neural networks. The three contributions are 1) measuring  intrinsic dimension of common image datasets (e.g., MNIST, CIFAR, ImageNet, COCO, CelebA), 2) demonstrating using GANs, for which one has control of the intrinsic dimension, the effectiveness of their dimension estimator for images and 3) tying intrinsic dimensionality to generalization performance.

Strength:

I am actually a bit surprised that no existing work to my knowledge has carefully measured the intrinsic dimension of modern image datasets. A work of such provides important justifications for numerous work on understanding and designing CNNs based on low-dimensional assumptions.  Therefore, I appreciate the novelty and the significance of the work very much.

Weaknesses:

My main concern is that the work appears underdeveloped in its current form and does not convincingly justify its conclusion. In particular, estimation of intrinsic dimension of dataset is the foundation for the development of this work, but the discussion for it is very shallow:

1. It is not clear why (1) provides an estimate of intrinsic dimensions. Under what assumptions is it derived? For example, is it assuming that the manifold is mostly flat or does it also work when manifold is curved?

2. There is no discussion on the effect of k and how it should be selected. What could go wrong if I pick k to be too large or low small? Will it result in over-estimation or under-estimation of the dimension?

3. How is the dimension of the subspace/manifold affects the choice of k? This question is relevant to Table 1 where the comparison of dimension for different datasets are based on the same k = 5, 10 or 20, but likely the best choice of k is different for different datasets. I'd also suggest an experiment in the GAN setting where such effect is demonstrated through experiments, perhaps with a figure similar to Fig. 3 but with a fixed number of samples and varying n in the x-axis.

4. Estimation of intrinsic dimension from data has been extensively studied (see e.g. [a, b, c]), and many recent works have used such notions to develop robust deep learning methods [d, e]. There should be a explanation on why (1) is picked over the other choices, e.g., are they better choices than those used in [d, e]?

a. Maximum Likelihood Estimation of Intrinsic Dimension, 2004
b. Estimating the intrinsic dimension of datasets by a minimal neighborhood information, 2017
c. Local Intrinsic Dimensionality I: An Extreme-Value-Theoretic Foundation for Similarity Applications, 2017
d. Dimensionality-Driven Learning with Noisy Labels, 2018
e. CHARACTERIZING ADVERSARIAL SUBSPACES USING LOCAL INTRINSIC DIMENSIONALITY, 2018

Overall, I am not fully convinced as to whether the estimation of the intrinsic dimension from this paper is a faithful and good enough characterization of the true intrinsic dimension. My suggestion is that the paper provides a review of existing dimension estimation methods,  clearly points out the pros and cons of each (conceptually and perhaps also by experiments), explains the reason for the specific choice in the paper, and provides some discussion on the properties of the specific dimension estimation method.

Minor comment:

- A key challenge in estimating intrinsic dimension of images in e.g. ImageNet is that they lie in relatively high-dimensional subspaces and suffer more from curse of dimensionality. Therefore, it may be beneficial to use data augmentation for generating more sample points, which may help to improve the precision of estimation.

- Fig. 3 is not color blind friendly, maybe consider using different line types.

- Sec. 5, first line: establish -> established


**Update after rebuttal**

I would like to thank the authors for the additional details provided in the rebuttal and revised version. All my concerns have been adequately addressed. Given the importance of the topic and the development is reasonably solid, I would like to recommend for its acceptance.

---

> ### Author Response · Authors · 2020-11-24
> **We appreciate your feedback.**
>
> Thank you for acknowledging the novelty and significance of our work. We appreciate your feedback.  We have addressed each of your comments below.
>
> 1.  Thank you for pointing this out.  We should have been more clear in the paper.  The original work on the MLE method contains validation on a variety of manifolds, and we did not want to replicate that work.  However, we did feel it was important to validate the methods on datasets with image structure, which is why we did a study that revealed that the method remains fairly accurate when applied to high-resolution images generated by a GAN trained on ImageNet.  We have added a discussion regarding assumptions and validation to the paper (cf. Section 3).
>
> 2.  It is argued by Levina and Bickel that, for large n and k the MLE estimator is unbiased. However, in the finite sample case, there is a tradeoff between bias and variance.  For small k, the estimator has low bias but high variance, whereas large k yields a high positive bias.  The increasing bias is consistent with our findings from Figures 3, 9, 10, 11, and 13.  In these figures, large values of k generally results in higher estimates, and in some cases, these values demonstrably yield overestimates given our controlled GAN setup.
>
> 3.  Thank you for the suggestion.  We have now added a table containing this information to our current draft in Appendix A.4 (Table 5).  From Table 5, a smaller value of k tends to yield a better estimate, especially when the intrinsic dimension is not very large.  Note that the number of free entries in the GAN’s latent vector only provides an upper bound for the ID of the produced images.  While we expect that this upper bound is close to the true dimension for smaller numbers of free entries, we do not have reason to believe this is true for very high numbers of free entries.  We have updated our draft to perform all experiments with multiple values of k, and we the trends we observe do not depend on which k is chosen.
>
> 4.  We agree that a deeper discussion of existing dimension estimation methods is warranted, and we have updated our current draft to reflect this in Section 3 and Appendix A.5.  Of the sources you suggested, [a] proposes MLE, the original formulation of the estimator we use, and [d, e] (equation 4 in both) also use this same estimator.  [b] assumes uniformly distributed data on a manifold which is not realistic for natural images, and this estimator also requires computing all pairwise distances (not tractable for our large-scale studies where the number of samples is as large as 2M).  We also did not extensively discuss the theoretical foundations of local intrinsic dimensionality like [c], but MLE is one of the local intrinsic dimensionality estimators that has the best trade-off between statistical efficiency and complexity.  Additionally, we have now provided validation experiments on several other estimators we tried including GeoMLE, TwoNN, and the kNN graph distance method in Appendix A.5.  We first validated that each of these methods are effective on hypercube data with uniform distribution, and then we compared them to MLE on GAN data and popular datasets, finding that MLE consistently yields more accurate estimates.
>
> Regarding the “minor comments”, thank you for pointing these out.  We have made the associated edits.

---

### Official Review · AnonReviewer1 · 2020-11-05
**Tackling an interesting question, experiments could be improved**

**Rating:** 7
**Confidence:** 4

**Review:**

== Summary ==

The paper studies the relationship between the intrinsic dimension of images and sample complexity and generalization. The authors suggest to use a variant of the MLE method of Levina & Bickel (2004), which is based on computing the distances to nearest neighbors in pixel space, which is fairly easy to implement and

== Pros ==

- The authors aim to investigate two relevant hypotheses for the field of representation learning. 1) intrinsic dimension of images is much lower than extrinsic dimension, and 2) extrinsic dimension has little effect on sample complexity.

- To test hypothesis 1), they use an estimator of the intrinsic dimension, and measure its fitness in a controlled setting for which the know the real intrinsic dimension (images generated by an state-of-the-art GAN). Hypothesis 1) is confirmed under this controlled setting and under a real scenario.

- Section 5.1. shows the (inverse) correlation between intrinsic dimension and sample complexity, and shows that extrinsic dimension (i.e. number of pixels) has a much weaker correlation. This section aims to confirm hypothesis number 2).

- I also find interesting the experiments in section 5.2, which studies the (inverse) correlation between intrinsic dimension and generalization (i.e. test accuracy).

== Cons ==

- Caption in Figure 3 states that the authors "observe the estimates to converge around the expected dimensionality of 10". However, the dimensionality estimate greatly depends on k, the number of neighbours used for each image. No variance/confidence interval methods are reported, in this figure, so it's unclear whether the differences between 12 and 10 are large or not (although they seem small if one compares against the extrinsic dimension of the images: 128x128x3).

- This paper uses yet another intrinsic dimension estimator, different from Gong et al. 2019 and Ansuini et al. 2019. It's unclear what's the impact of the estimator in the predicted value of the intrinsic dimension.

- One of the emphasized contributions of the paper is that it's "the first to show that intrinsic but not extrinsic dimensionality  matters for the generalization of deep networks" (page 6). As far as this reviewer is aware, indeed this paper is the first to measure intrinsic dimensionality *of images* and its relationship with generalization, but there are others that compare the intrinsic dimensionality of the final embedding with accuracy, showing the same conclusion (e.g. Gong et al. 2019, Ansuini et al 2019). Thus, the authors should be more specific when talking about intrinsic/extrinsic dimensionality (it refers to the image, not the embedding representation of a given deep neural network classifier).

- Both the intrinsic dimensionality of the images and the classifier will impact the accuracy. This paper only focuses on the former, while other papers focus on the latter. Since this paper is posterior to the aforementioned papers, it would be appreciated if the authors could comment on which intrinsic dimensionality shows larger correlation with generalization, and draw some relationship among them.

- Some figures can be hardly read if printed in grayscale (Figures 3, 6, 7, 8). I would suggest to use different line styles to better discern among curves in the plots, and using hatches in the histograms (Figure 1).

- In the introduction, it seems that the authors missed important seminal works on autoencoders (e.g. "Reducing the dimensionality of data with neural networks" by Hinton and Salakhutdinov, 2016), since their references for autoencoders and regularization methods date back only to 2018.

- I have some minor concerns regarding computational cost. The authors use a fraction of images as "anchors" and compute the nearest neighbour against the rest of the images in the dataset. This still leads to a quadratic cost in the number of images in the dataset, which may become problematic with modern datasets (ImageNet or even bigger ones). Given that the dimensionality estimates don't change much (e.g. Figure 3), why not fixing the number of samples to a constant number (e.g. 1000)?

== Rationale for the score ==

Although I raised many points in my "Cons" section, many of these are more questions rather than specific issues that I have with the presented paper. The paper tackles an important question of interest for the ICLR community: how to estimate the intrinsic dimensionality of our datsets, and which impact does it have on generalization and sample complexity, and it does so with a quite convincing experimental setup. The method proposed by the authors could have important applications, such as estimating the number of required training samples for reaching a target accuracy.

I hope that the authors can address my questions/concerns during the rebuttal to increase my score.

*Update after discussion*: The authors have addressed all the points that I raised during the discussion. I appreciate the effort, and I'm increasing my score accordingly.

---

> ### Author Response · Authors · 2020-11-24
> **Thank you for your insight.**
>
> Thank you for your insight.  We address your comments in order below:
>
> *“variance/confidence interval methods”*
>
> Thank you for the suggestion.  We have now added error bars to Figure 3.
>
> ---
>
> *“what's the impact of the estimator in the predicted value of the intrinsic dimension”*
>
> Our paper uses the previously validated MLE method of MacKay & Ghahramani, however it is certainly true that other methods exist and it would be very informative to compare across them.  We have now added results for the GeoMLE, TwoNN (used by Ansuini et al. 2019), and kNN graph distance (used by Gong et al. 2019) methods to the updated draft (see Appendix A.5), and we observe that these other dimension estimators do not perform as well as MLE on image data.  Note that we did validate the MLE method in controlled settings with synthetic images, and we run many of our learning experiments on this same synthetic image data.  Although TwoNN and kNN graph distance methods are shown to work well on image features, we find that they are not reliable on images (see Figure 14 and Figure 15).
>
> ---
>
> *“more specific when talking about intrinsic/extrinsic dimensionality ”*
>
> At your suggestion, we have now clarified this distinction in our draft and made clear that other papers have made substantial contributions to the study of dimensionality in internal representations.
>
> ---
>
> *“which intrinsic dimensionality shows larger correlation with generalization, and draw some relationship among them.”*
>
> This question is thought provoking and difficult to answer.  The dimensionality of the feature representation is bounded above by the data dimension, and there may be a further and interesting relationship.  We do not want to speculate about this relationship and its cause, but this would be an interesting direction for future work.
>
> ---
>
> *“use different line styles… hatches in the histograms”*
>
> Thank you for the formatting suggestions.  We have made the suggested changes.
>
> ---
>
> *“seminal works on autoencoders ”*
>
> Thanks for pointing this out.  It was an oversight to omit well-known works on autoencoders, and we have now cited two more seminal papers in the introduction.
>
> ---
>
> *“quadratic cost in the number of images in the dataset”*
>
> It’s true that the scaling is quadratic, but in practice, the constant-factor speedup we get from sub-sampling was sufficient to achieve tractability, and so we did not have a need for linear scaling, but your point is well taken.

---

### Decision · Program_Chairs · 2021-01-07
**Final Decision**

**Decision:**

Accept (Spotlight)

**Comment:**

There was a consensus among reviewers that this paper should be accepted as the authors addressed reviewers' concerns in the discussion phase. This paper is well-written and easy to read. It provides a coherent story and investigation on two important hypotheses: that natural images have a lower intrinsic dimension than the extrinsic dimension (e.g. the number of pixels) and that a lower intrinsic dimension lowers the sample complexity of learning. These results appear to be novel and significant for the ICLR community as it provides justifications for numerous work on understanding and designing convolutional neural networks based on low-dimensional assumptions.